# Molecular Mechanisms in Murine Syngeneic Leukemia Stem Cells

**DOI:** 10.3390/cancers15030720

**Published:** 2023-01-24

**Authors:** Michael Chamo, Omri Koren, Oron Goldstein, Nir Bujanover, Nurit Keinan, Ye’ela Scharff, Roi Gazit

**Affiliations:** The Shraga Segal Department of Microbiology, Immunology and Genetics, Faculty of Health Science, Ben-Gurion University of the Negev, Beer Shevap 8410501, Israel

**Keywords:** syngeneic model, AML, leukemic stem cell

## Abstract

**Simple Summary:**

Cancer treatments usually gain good responses; however, some tumors relapse frequently. Acute Myeloid Leukemia (AML) is notorious for its robust relapse. This is attributed to the leukemic stem cells (LSCs). We used a murine syngeneic leukemia model, ML23, to identify and study LSCs in syngeneic settings. Hereby, we present the prospective isolation of a defined LSC sub-population, encompassing the potency to pass disease from mouse to mouse. We further provide molecular insights and whole transcriptome analysis. Importantly, the ML23 LSC sub-population expresses therapeutic targeted genes and provides a model for research in immune-competent animals.

**Abstract:**

Acute Myeloid Leukemia (AML) is a severe disease with a very high relapse rate. AML relapse may be attributable to leukemic stem cells (LSC). Notably, the “cancer stem cell” theory, which relates to LSCs, is controversial and criticized due to the technical peculiarities of the xenotransplant of human cells into mice. In this study, we searched for possible LSCs in an immunocompetent synergetic mice model. First, we found phenotypic heterogeneity in the ML23 leukemia line. We prospectively isolated a sub-population using the surface markers cKit^+^CD9^−^CD48^+^Mac1^−/low^, which have the potency to relapse the disease. Importantly, this sub-population can pass in syngeneic hosts and retrieve the heterogeneity of the parental ML23 leukemia line. The LSC sub-population resides in various organs. We present a unique gene expression signature of the LSC in the ML23 model compared to the other sub-populations. Interestingly, the ML23 LSC sub-population expresses therapeutic targeted genes such as CD47 and CD93. Taken together, we present the identification and molecular characterization of LSCs in a syngeneic murine model.

## 1. Introduction

### 1.1. Acute Myeloid Leukemia (AML)

Leukemia is a severe life-threatening disease that affects children and adults. Patients suffer an increased proliferation of leukocytes and the inhibition of normal blood cell production [1]. Acute Myeloid Leukemia (AML) is a malignancy of the myeloid progenitor cells [2]. It is prevalent in the elderly, and the five-year survival rate of patients >60 years is less than 15% [3]. In this process, bone marrow (BM), blood, and other tissues are infiltrated by malignant leukemic cells [4]. The classification of the types of AML used to relay phenotypes have moved to genetics in recent years [5]. Genetic alteration of the malignant cells hampers the differentiation of erythrocytes, platelets, monocytes, and granulocytes [6]. AML patients also suffer aberrations of their BM, causing insufficient hematopoiesis and immune activities. Garry Gilliand et al. suggested that most AMLs are driven either by mutations that activate signal transduction pathways, thereby enhancing proliferation or survival of progenitors, or by mutations effecting transcription factors (TF), such as the HOX genes, hindering normal hematopoietic differentiation and enhancing self-renewal [7,8]. In a previous work, we presented the leukemia model for AML, ML23, which is similar to the second type, using the over expression of *Meis1*, *HoxA9*, and *HoxB5* [9]. Interestingly, mouse models can recapitulate human AML, allowing for functional studies, drug development, and detailed molecular studies [10].

### 1.2. Leukemic Stem Cells

Previous works on AML in a murine xenograft model have reported the prospective isolation of a sub-population of cancer cells that can pass AML [11]. These cells, defined as leukemia-initiating cells (LICs), possess some similarities with hematopoietic stem cells (HSCs), such as multipotency and self-renewal characteristics [12]; therefore, they are also referred to as leukemia stem cells (LSCs). Later studies have reported the identification of sub-populations that bear the engraftment potency of other cancers, which are sometimes referred to as “cancer stem cells” (CSCs) [13,14,15,16]. AML often presents a cellular hierarchy that may resemble normal hematopoiesis [14]. Malignancy is preserved using LSCs; moreover, while differentiated cells may encompass a large portion of a cancerous mass, they lack the ability to relapse [17]. Importantly, new therapeutics are tested to target the dominant blast population of leukemia; however, researchers might not test for the targeting of the LSCs [18,19]. Moreover, localization of LSCs in supporting niches, and their relatively slow cycling, may protect them from some of the common and new treatments [19,20]. AMLs are notorious for relapsing [3,19,21]. Identification, functional examination, and molecular study of presumable LSCs in additional leukemia models will enable the development of better treatments [3,22,23].

### 1.3. Murine Models for AML

Mouse models have been leading cellular and molecular leukemia research for over 70 years [1,24]. From carcinogen-induced leukemia [25], random transgenics, genetically engineered mice [26], and xenotransplantation—each has advantages and limitations [10,27,28,29]. Importantly, animal models allow for better experiments and pioneered the prospective identification of LSCs [10,11]. Many molecular and cellular discoveries were possible only thanks to mice models, while each model is truly limited. Importantly, a major criticism of the “cancer stem cell” theory was supported by some mice models [30]. Strasser et al. demonstrated an extremely high frequency of leukemic cells that robustly pass the disease, disputing the concept of a minor LSC sub-population [30]. Interestingly, the xenotransplant of human leukemia into mice was initially recognized as requiring a supplementation of human cytokines [11], and the multiple interactions of malignant cells with their niche are not fully recapitulated, casting doubt on LSC identification [11,31] across species. Moreover, xenotransplantation requires the host to be immune deficient; however, currently, there is a high interest in immune–leukemia interactions. Therefore, additional independent models of leukemia in syngeneic mice are of interest.

### 1.4. A Novel Murine Model of AML LSC

In this study, we utilized an AML-induced model with the transgenic expression of defined oncogenes [9]. First, we realized the heterogeneity of surface markers on the ML23 leukemia line. Then, we fractionated the malignant cells into sub-populations and tested their ability to pass the disease. Importantly, we identified one defined sub-population that encompasses the potency of the passage and found it to fully reconstitute the heterogeneity of the original ML23 leukemia line. This fulfills the practical definition of LSCs, shown here in a syngeneic mice model. Analysis of whole transcriptome data identifies a unique set of differentially expressed genes, including additional markers which allow for validation. Finally, we found differential expression of several targets for treatments that may provide novel ways to better treat AML and hopefully eradicate the disease.

## 2. Materials and Methods

### 2.1. Animals

Mice were kept in the specific pathogen-free (SPF) unit at the Ben-Gurion University; all experiments were performed according to ethical committee approval and under local and state regulations, protocols IL01-01-2017D and IL29-05-2021C. Leukemia lines were previously described [9].

### 2.2. Blood Sampling and FACS Processing

Mice were bled into 150 µL of Alsever’s solution. Samples were then treated with 10 mL ACK buffer for 2 min and centrifuged at 1600 RPM for 5 min. The supernatant was removed, and pellets were washed in 5 mL sample media (SM, PBS with 2 mM EDTA and 2% FCS). Antibodies of Biolegend (San Diego, CA, USA): CD45.1 APC, CD45.2 pacific blue, Mac1 PE-Cy7, B220 APC-Cy7, CD3e PE, Lineage-PacificBlue (Including- anti Ter119, Mac1, Gr1, CD3e, CD4, CD8 and B220), c-Kit Alexa780, Sca1-APC, CD150-PEcy7, CD48- Percp cy5.5, CD9-APC, CD34-PE, CD84-PE. Cells were stained on ice for 1 hr and washed. Flow cytometry analysis of the reporter expression frequencies (ZsGreen+) and the surface markers’ expression were performed on the Gallios Flow Cytometer (Beckman Coulter, Brea, CA, USA). Fluorescence activated cell sorting (FACS) data analysis was performed using Kaluza v1.2 (Beckman Coulter). Sorting was performed on FACSAria III (BD), as previously reported [9].

### 2.3. RNA Sequencing and Analysis

Following the isolation of sub-populations, we used the NEXTflex Rapid Directional RNA-Seq Kit (catalog NOVA-5138-01) for library construction; then, we used the Qubit DS DNA chip for quality assurance. The G-INCPM facility performed the RNA-sequencing, yielding 5–15 million single-reads of 61 bases per sample. Data were analyzed using Partek Genomics Suite (Partek, Inc., St. Louis, MO, USA), briefly: alignment was carried out with STAR v2.5.3a (reference index: mm10–Ensembl Transcripts release 92), quantification with the Partek algorithm (Quantify to annotation model (Partek E/M)) followed by normalization and differential expression analysis DEseq2 v3.5. For principal component analysis (PCA), reads were converted to Log of base 2, we used cutoff 6 in at least one sample, and Partek PCA by variance. Annotated GO gene lists were obtained from MGI and included in the Appendix A.

## 3. Results

### 3.1. ML23 Leukemia Line Can Passage Serially and Present Heterogeneity

We have previously managed to produce murine AML-like cells through the overexpression of 3 oncogenes: *Meis1*, *HoxA9*, and *HoxB5* [9]. Bone marrow or spleen cells were extracted from diseased mice and proved potent for passage into secondary and tertiary recipients, as previously reported (Figure 1a, similar to published [9], independent experiments). The development of leukemic cells was monitored by measuring the expression of the fluorescent reporter ZsGreen+ in peripheral blood (PB) samples. We found a small percentage of cells with a high expression of ZsGreen+ 3 and 4 weeks after the passage (Appendix A: ML23 line presents consistent and stable leukemia development). Therefore, we could determine the propagation of leukemic cells derived from ML23 transplanted cells. We chose 12 surface markers that characterize HSCs (cKit, CD150), progenitor cells (Flk2, CD48, Tie2, CD9), or differentiated cells (CD16/32, Gr1, Esam1, Mac1, CD11c, F4/80). While not all of the markers show heterogeneous expression, Mac1, CD9, cKit, CD48, F4/80, GR1, and CD11 did (Figure 1b, extending panel of previous publication [9]). Similar to AML cells, the ML23 cells contain heterogeneity among the leukemia cells. This is demonstrated by the various cell markers and their expression, with some populations having the ability to spread the disease [11]. Thus, ML23 can effectively pass and present with the heterogeneity of surface markers.

### 3.2. Prospective Isolation of LSC in the ML23 Leukemia Line

We sought to sort the ML23 cells into sub-populations and assay their ability to transfer the disease (Figure 2a). We chose four markers that showed heterogeneous CD9, cKit, CD48, and Mac1(Figure 1b). CD48 and cKit were described as HSC markers and revealed a heterogenous expression within the ML23 cells [9]. Mac1 serves as a marker for the myeloid lineage cells [32]. CD9 serves as a marker for leukemic cells and HSCs [33,34]. These four markers enable us to separate the ML23 cells into distinctive sub-populations (Figure 2b). The sort was of ZsGreen + cells; defined population were P9 = CD9 + cKit + Mac1 + CD48-, P10 = CD9-cKit + Mac1 + CD48+, P11 = CD9-cKit + Mac1-CD48+, P12 = CD9-cKit-Mac1-CD48+. We transplanted each sub-population into four different groups of mice, while an equal number of cells were transplanted into each mouse. Mice that received P11 cells developed increasing amounts of ZsGreen+ cells in PB. In contrast, none of the other groups of mice (P9, P10, P12) developed the disease, as we saw only a little or usually no expression of the reporter ZsGreen+ from these groups (Figure 2c and Appendix A: FACS plots for ZsGreenin the different subpopulations of ML23). Mice from sub-population P11 also developed enlarged lymph nodes, visibly white bones, and an enlarged spleen. These physiological characteristics are common features of leukemia [4,35,36]. In contrast, none of the other groups portrayed these characteristics (Figure 2d).

### 3.3. The P11 LSC Sub-Population Reconstitutes the Heterogeneity of ML23

After establishing that sub-population P11 has the potential to relapse disease, we wanted to examine if it reconstituted only itself or perhaps additional sub-populations of the parental ML23 line. We extracted cells from the group of mice injected with the P11 sub-population (Figure 3a) and stained for the four surface markers. Interestingly, all four sub-populations (P9, P10, P11, and P12, as defined above, for short P9 CD9+, P10 cKit+Mac1+, P11 cKit+Mac1-, P12 cKit-) were present in the primary mice transplanted with P11 sorted-cells (Figure 3b). Hence, mice transplanted with P11-gated cells (Figure 3a) developed malignant cells with a heterogenous phenotype (Figure 3b). Moreover, the relapsed heterogeneity showed a cell distribution pattern which resembles the original ML23 leukemia. Thus, the P11 sub-population presents both self-renewal characteristics and the potency to reconstitute the heterogeneity of the original ML23 leukemia. Accordingly, P11 is a bona fide LSC sub-population.

### 3.4. ML23 Sub-Populations Distribute in Various Organs

LSCs are suggested in some models to localize in the BM [37]. We sought to examine the distribution of the sub-populations of ML23 leukemic cells in the syngeneic ML23 model. A previous study suggested that the spleen contains high amounts of leukemic cells and shares the same engraftment as cells from the BM [38]. Thus, we examined BM (femur, tibia, and pelvis), spleen, lymph nodes (LN), and thymus. In all examined organs, large amounts of the leukemic ML23 cells that comprised the four sub-populations were evident at advanced stages of the disease. We had no specific reason to suspect major differences between the BM of the femur, tibia, and pelvis; indeed, these showed a similar composition of the sub-populations (Figure 4). Interestingly, we found a similar distribution pattern of the sub-populations in the organs; the largest sub-population was P12, whereas the rarest sub-population was P9. We also found that, unlike other organs, the spleen had higher frequencies of the sub-population P10 (Figure 4 and Appendix A: Distribution of the sub-populations of ML23 in various organ). Importantly, we found that the LSC sub-population, P11, was observed in all organs in a similar relative amount, except for the spleen, where it was expressed in lower levels relative to the other organs (Figure 4). Hence, in this syngeneic ML23 model, phenotypic LSCs distribute to the multiple organs examined w/o significant restriction to the BM.

### 3.5. The Unique Gene Expression Signature of LSC

Next, we examined the transcriptome of the sub-populations of the ML23 leukemia line. We harvested cells from three diseased mice and sorted them into the four sub-populations for RNA-seq, as described in Materials and Methods. We defined statistical cutoff, which considered the average expression of each gene from the non-LSC sub-populations compared to the average expression in the LSC (P11), the false discovery rate was (FDR < 0.1, fold change > 2 or <−2). Out of 20,020 genes, we found 470 significantly and substantially downregulated or upregulated (Figure 5a, Appendix A). PCA found that P12 appears to be closest to P11, while P9 is the farthest sub-population (Figure 5b). We also found that the P12 cells shared a similar gene expression profile to P11 compared to P9 and P10, as can be seen in the uniqueness of the expressed genes in P11, which is partially shared with P12 (Figure 5c). Thus, P12 is the closest population to P11.

To validate our data, we first examined the RNA expression of the four cell surface markers (CD9, cKit, CD48, and Mac1) to determine if mRNA expression correlated with surface proteins from which cells were sorted (Figure 3). Clear agreement was seen for all mRNA expressions with the cell surface markers’ prevalence within the four sub-populations (Figure 5d). We can see the high expression of *Cd9* mRNA only in P9; *Cd48* is highly expressed in all the sub-populations except for P9 (Figure 5d). These findings also match the expression levels of the two subunits of the surface marker protein Mac1 (*Itgam*, *Itgb2*), where both P9 and P10 showed expression; however, P11 and P12 lacked it (Figure 5c). Although surface expression levels and mRNA expression levels do not have to match because there are several regulation processes between mRNA transcription and protein synthesis, our findings validate the data and ensure that no error was made from the processing of the cells to data analysis.

### 3.6. Additional Surface Markers Validate Differential Expression on LSCs vs. Other ML23 Sub-Populations

Out of the specific LSC-expressed genes, we focused on a few surface markers. Some 85 markers are upregulated or downregulated in P11 (Appendix A: LSC has a unique surface markers pattern: Appendix A). We chose two additional surface markers: CD34 and CD84 (Figure 6a). These markers were reported to be overexpressed [11] or underexpressed [39] in other LSCs. Fresh cells were collected and stained for the four initial surface markers in addition to CD34 and CD84, allowing for the identification of the expression on the previously defined sub-populations. We found clear staining for CD34, showing higher expression on the P11 and P12 cells; on the other hand, CD84 showed higher levels in the P9 and P10 cells (Figure 6b, c). Thus, additional surface markers selected from the RNA-seq data show the same trend of protein expression at the surface of the cells, further validating our data and suggesting good predictive value from RNA to proteins.

### 3.7. Specific Drug-Targeted Genes Expressed in ML23 Cells

More than a dozen unique LSC immunophenotypic components were reported for the drugs that were developed [40]: CD123/IL-3 receptor [41,42], CD44 [43], CD33 [44], CD47/SIRPα receptors [45,46], CD96 [47], CD93 [48], CD25, CD32 [49], Tim-3 [50], CD99 [51], and IL1-RAP [52]. We examined the differential expression levels of these clinically relevant drug target proteins on our ML23 LSCs, as well as on P11 versus the rest of the leukemic cells (P9, P10, and P12). We found that CD47 and CD93 are overexpressed in the P11 sub-population compared to the other sub-populations of ML23 (Figure 6d). Thus, our data demonstrate the possible preferential targeting of LSCs in this model using clinically relevant surface markers. The identification of differentially expressed drug-targets may allow for the elimination of more potent cells that relapse disease.

## 4. Discussion

### 4.1. Overview

In this study, we sought to identify LSCs in the ML23 murine leukemia model. ML23 leukemic cells effectively pass in syngeneic C57Bl/6 mice and present with a heterogeneity of 7 surface markers out of the 12 examined. We sorted four sub-populations for functional testing and found one, P11, to encompass robust passage potency. P11 not only passaged the disease but also retrieved ML23 heterogeneity. We report the unique transcriptome signature compared to the other sub-populations, with 470 defined genes. Additional surface markers, out of these 470 genes, validate our data. Finally, we show the differential expression of new AML targets, including the preferential expression of CD47 and CD93 on the ML23 LSCs.

### 4.2. Heterogeneity of ML23 Cells

Previous works on murine models for AML, such as the MLL-AF9 model, have presented the heterogeneity of the leukemic cells. Heterogeneity was reported for cKit and CD24 [53,54]. Similar to the work of Saadatpour A. et al., 2014, our ML23 model presented a heterogenous expression of the surface markers Mac1, CD9, cKit, CD48, F4/80, GR1, and CD11c [53]. We selected four main markers to separate four sub-populations, namely gated as P9-P12. As reviewed by Pollyea and Jordan, AML hierarchy entails a process in which a single cell forms a heterogeneous lineage. Intriguingly, our model, similarly to human AML, can be inherently heterogeneous [40,55].

### 4.3. Prospective Isolation of LSCs and the Reconstitution of the Disease

ML23 heterogeneity enabled us to prospectively isolate four sub-populations. The “cancer stem cell” theory is under controversy, with major interest in better understanding and eradicating malignant diseases [30,56]. LSCs were first described in a xenotransplant model [11,12]. Xenogeneic models benefit the direct study of primary human patient cells. However, some human and murine cytokines and ligands–receptor interactions are incompatible [27]. Xenotransplantation also requires a severe immune deficient host, limiting an important aspect of leukemia study and treatment [10,28,29]. In contrast, syngeneic models benefit from perfect compatibility of the leukemic cells and the host, with no immune deficiencies [29]. In this research, we identified the sub-population P11 as bona fide LSCs. The P11 sub-population passaged disease and retrieved the heterogeneity of the ML23 leukemia line, similar to human AML LSCs [57]. Interestingly, unlike the MLL-AF9 in which LSCs were reported as Mac1-high [36], ML23 P11 LSCs are Mac1-low. Clearly, not all human leukemias have defined LSCs. Importantly, not all murine AML models have LSCs, as reported by Strasser et al. [30], driving the controversy over the CSC theory. Hereby, we report the identification of LSCs in the ML23 syngeneic model. Additional syngeneic models, having no incompatibilities limitations, will better determine which types of leukemia may follow CSC theory and which do not.

### 4.4. Distribution of Leukemia and LSC

The distribution of the ML23 sub-populations among different tissues demonstrates an advanced stage, as the leukemic cells were harvested from the mice when the ZsGreen+ reporter was close to 90% in the PB (Figure 4). Leukemia development in mice from ML23 can be detected early by identifying a small percentage of ZsGreen+ cells, while no other phenotypic evidence of AML is presented. This is in agreement with human AML progression, which is accompanied by the collapse of the BM niche by changes in the vascular structure and ends with osteoblast loss [58,59,60]. AML cells were shown to utilize the BM signals and catalyze the process of niche remodeling and adaptation to leukemogenesis, therefore impairing normal hematopoiesis [61]. Interestingly, hampering the connection between LSCs and the BM niche and promoting the reconstitution of normal HSCs has become a goal for new treatments [61]; one may test such novel treatments in the ML23 model. Moreover, the xenotransplantation of human AML into immune-deficient mice suggests a restricted localization of LSCs to the BM in some cases [62]; thus, studying syngeneic mice models is of high interest to gain a better understanding and broader applicability of new treatments. For example, Acute Lymphoblastic Leukemia (ALL) is known to reside not only in the BM, but also in the central nervous system [63]. Importantly, ALL can relapse after treatments and possesses an important target for novel CAR-T cell therapies [20]. Our findings suggest that, during the advanced disease stage, LSCs may spread across multiple organs and therefore require systematic treatments to be eradicated [64]. The capability of the ML23 cell line to serve as a model for early and for late stages allows it to contribute as a research tool for the critical assessment of treatments.

### 4.5. Molecular Signature of LSCs

RNA-seq was performed to confirm the uniqueness of the LSC sub-population, P11, by comparing it to the rest of the sub-populations. P11 showed a unique gene expression signature, which includes an extensive set of enriched or suppressed genes. Among the enriched genes in P11, we could identify genes with proliferative behavior or genes affiliated with an active cell cycle, such as *MYBBP1A* [65] (Appendix A). We also found a similarity in the gene signature between P11 and P12 that was different from P9 and P10. This information may indicate a possible hierarchical structure between the ML23 sub-populations. This is in agreement with previous studies, where heterogeneous AML cells demonstrated a hierarchy resembling the hematopoietic tree [18]. This hierarchy separates AML cells into groups according to their genomic, functional, and clinical properties, where it was shown that patients with a short survival suffered from enriched HSC-specific genes [18]. Therefore, as with human AML, our ML23 leukemia can present a hierarchy that can affect the outcome to certain treatments based on the composition and the stemness of the disease. Further research of this data may present a functional role to the other sub-populations or the interactions between them. From the RNA-seq of the surface markers CD34 and CD84, we found that CD34 is highly expressed in P11 and 12 but lacks the expression of CD84. These findings also correlate with the protein expression levels on the cell’s surface (Figure 6a–c). Expression levels of CD34 are in correspondence with the previous work of Lapidot et al., where LSC presented CD34+ expression [11]. Moreover, CD84- expression of P11 coincides with the idea of P11 as LSC, where CD84- expression was shown to be in less differentiated hematopoietic cells [39].

### 4.6. Possible Treatments to Eradicate Leukemia and LSCs

We compared the expression levels of genes that are up to date, proposed as LSC targeted therapy [40]. Among the ML23 sub-populations, we found markers (CD47 and CD93) that are upregulated in P11 and P12 and which also characterize LSC in the ML23 model (Figure 7). This is in agreement with previous studies on human LSC in AML, wherein CD47 and CD93 were expressed [45,46,47,48]. The conservation of a specific marker between mice and human is not trivial. However, when looking at LSC markers of AML, we must consider the subtype of AML and the variabilities among individuals [66]. Thus, we can see that our LSC, P11, showed to be beneficial as a model for CD47 and CD93 therapeutic research, while additional leukemia lines are needed to represent the pleura of AML subtypes fully.

## 5. Conclusions

In conclusion, our data provide a novel model for LSC in syngeneic mice. This ML23 and additional models where LSCs are present or not will improve our understanding of AML and other forms of tumor heterogeneity. Studying multiple syngeneic leukemic mice may resolve current controversies and realize variability relevant to the differences between human patients.

## Figures and Tables

**Figure 1 cancers-15-00720-f001:**
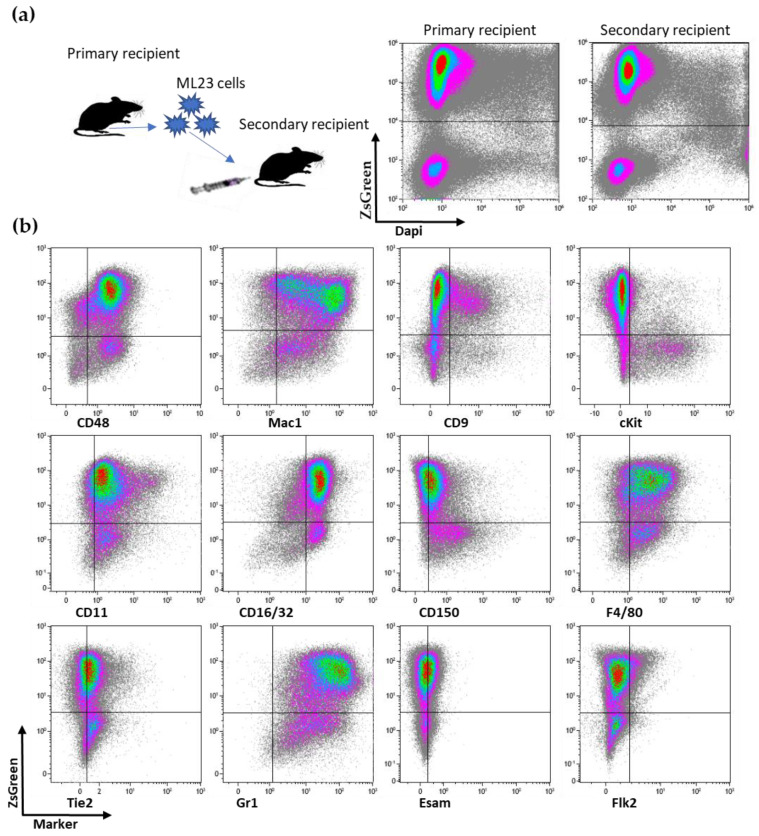
ML23 leukemia line passage serially with heterogeneous phenotype: (**a**) Mice were injected with leukemic ML23 cells from frozen stocks. After leukemia detection in PB, cells were harvested from the BM and injected into other recipients. FACS plot of the primary recipient is of one month after transplantation, and, for the secondary recipient, is from 2 to 3 wk after passage. (**b**) mFACS plots showing the expression of the indicated markers (*X* axis), along with the ZsGreen reporter (*Y* axis).

**Figure 2 cancers-15-00720-f002:**
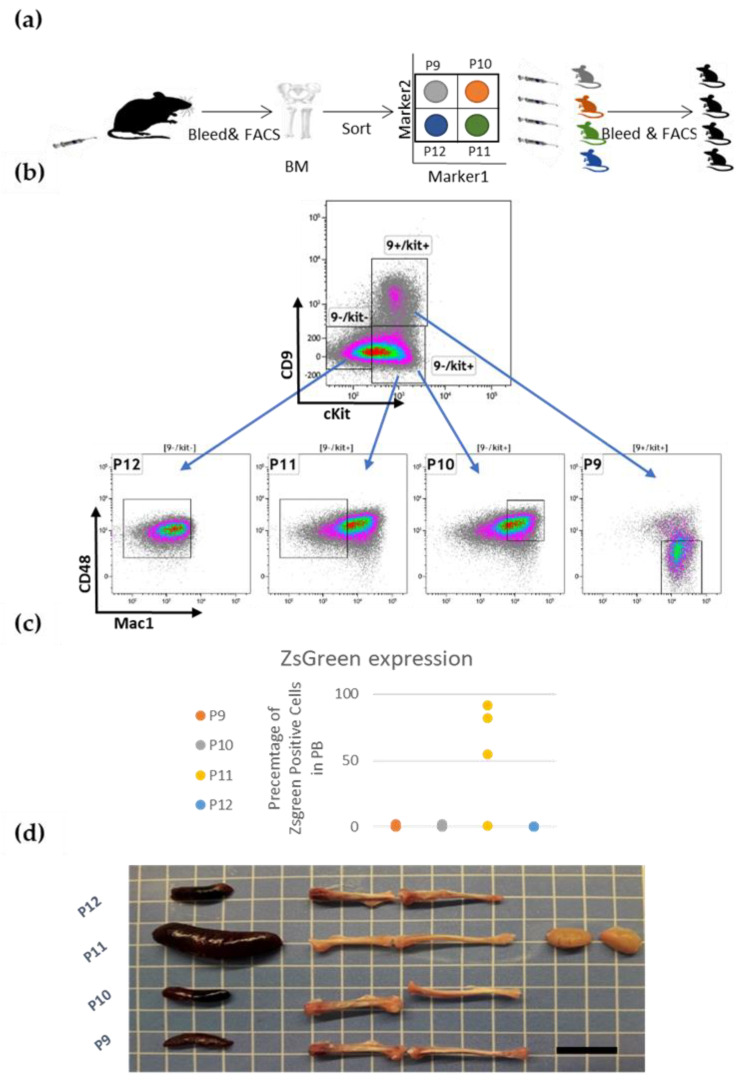
Prospective isolation of LSCs in ML23 (**a**) schematic illustration of experimental design. Fresh Leukemic cells were sorted into four sub-populations, and each sub-population was transferred into a new host and its effects were examined. Gating for ML23 sub-populations using cKit/CD9 and CD48/Mac1. (**b**) FACS plots of leukemic cells, pre-gated ZsGreen+, as sorted into defined sub-populations. (**c**) ZsGreen+ expression levels from each transplanted sub-population. P11 sub-population transplanted mice presented high ZsGreen+ expression; others did not. (**d**) Organs from animals that received the indicated sub-population of ML23. Enlarged spleen, white bones, and enlarged LN are shown from P11. Representative data are shown from one out of three independent experiments.

**Figure 3 cancers-15-00720-f003:**
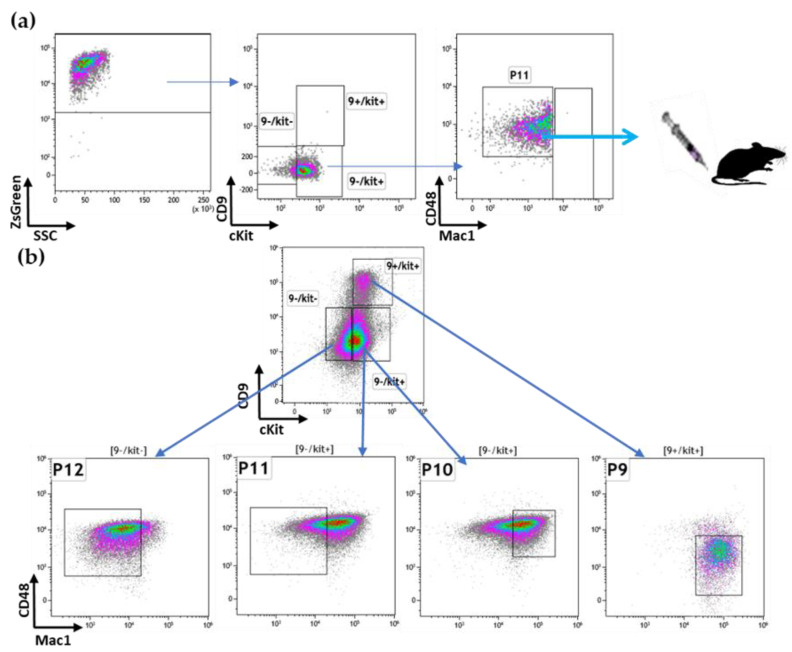
P11 LSC can reconstitute the heterogeneity of ML23: (**a**) Sorted P11 sub-population (CD9-cKit+Mac1-CD48+), which were transplanted into the recipient mice. (**b**) P11 sub-population reconstituted the heterogeneity of ML23 in the recipient mice (the primary recipient of P11-sorted cells). Representative data are shown from one out of three independent experiments.

**Figure 4 cancers-15-00720-f004:**
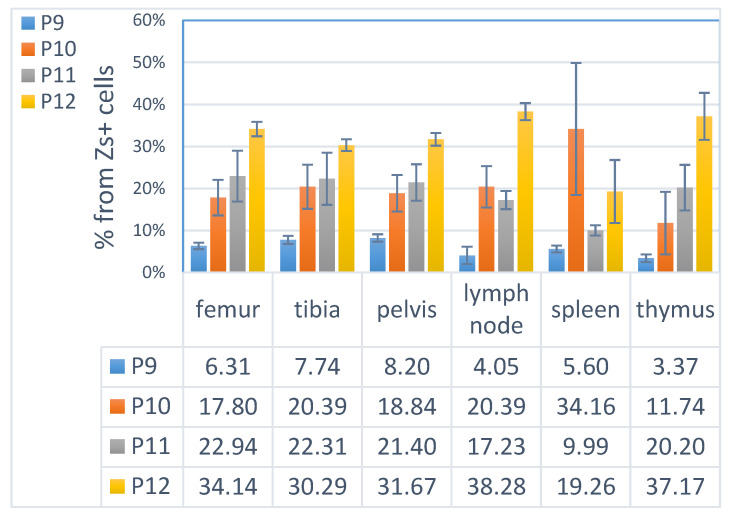
The sub-populations of ML23 distribute broadly in various organs. The average percentage expression (from ZsGreen+ cells) of each of the four sub-populations (P9—12) as measured using FACS analysis for cells extracted from six different tissues of ML23 leukemic mice. Cells were PRE-GATED to ZsGreen+, at least 90% in each sample (not shown). Data are shown from three independent experiments.

**Figure 5 cancers-15-00720-f005:**
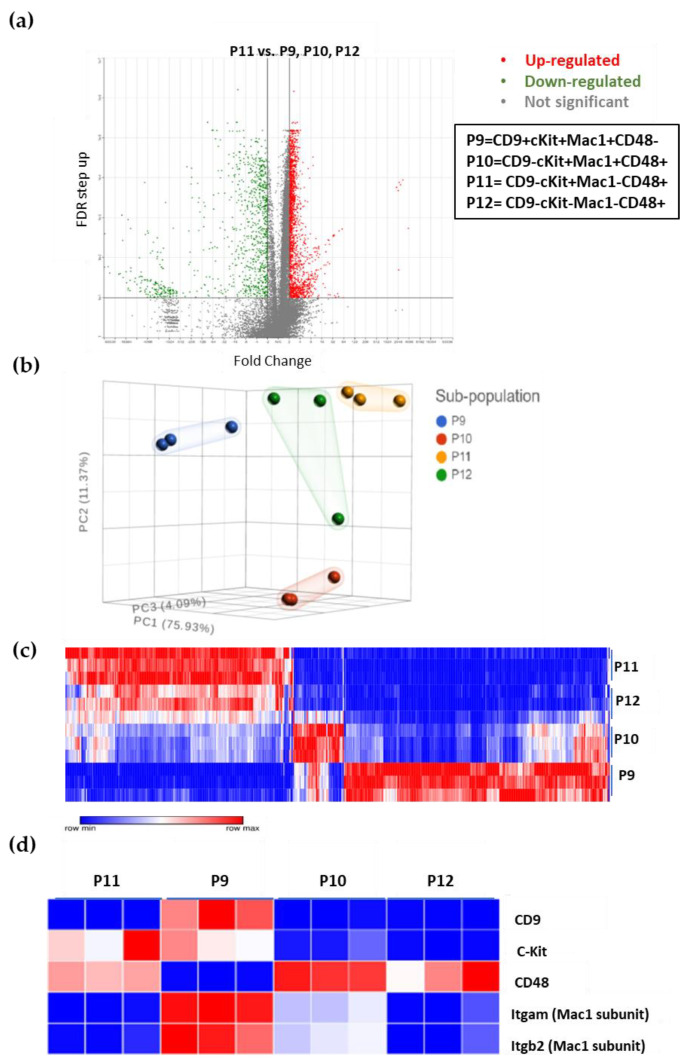
LSCs presenting a unique gene expression signature: (**a**) Some 470 genes (out of 20,020) are differentially expressed between the LSC sub-population (P11) and the other populations (FDR <0.1, fold change >2 or <−2). (**b**) PCA showing the 4 sub-populations (P9—P12). (**c**) Heatmap showing the 470 differentially expressed genes. (**d**) RNA expression signature of the four surface markers was found close to its expression at the protein level and used for validation (compare with Figure 3).

**Figure 6 cancers-15-00720-f006:**
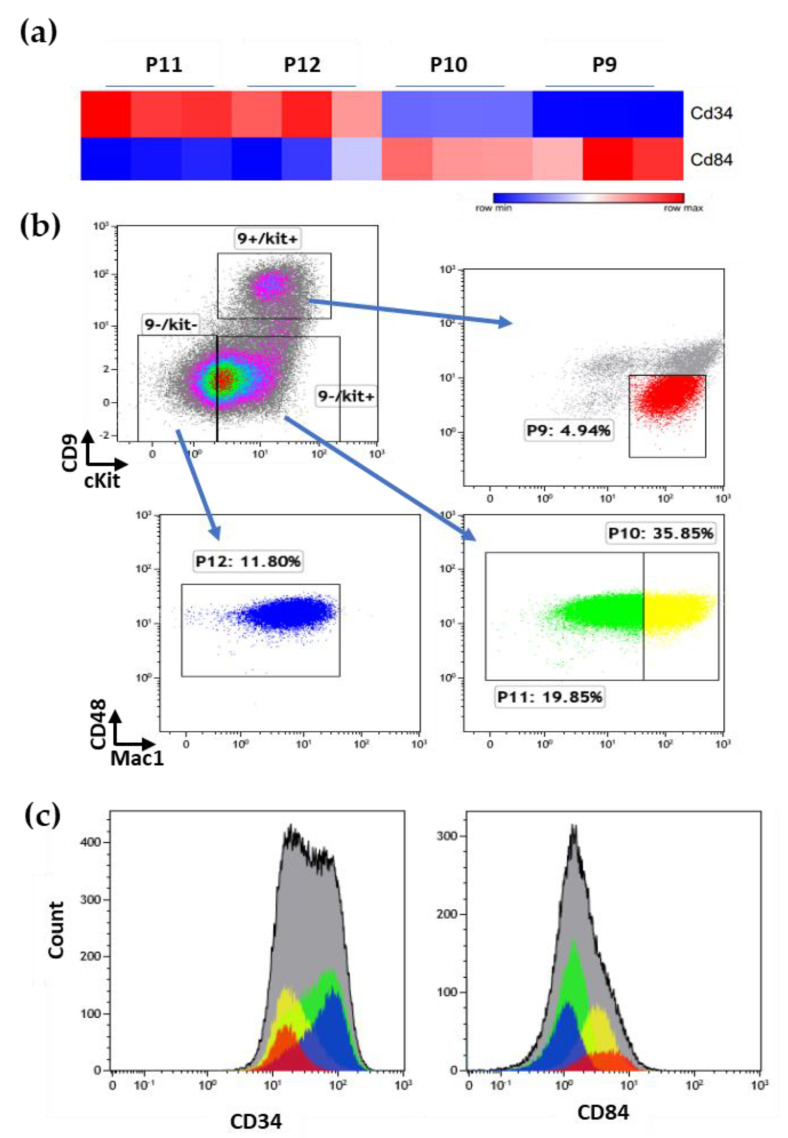
CD34 and CD84 are differentially expressed on ML23 sub-populations, in agreement with RNAseq data: (**a**) Heatmap of mRNA levels of *Cd34* and *Cd84* in the ML23 sub-populations. (**b**,**c**) Fresh ML23 cells were stained for the primary four surface markers in addition to CD34 and CD84 proteins. The expression level of each of the markers was examined in each of the sub-populations. Representative data are shown from one out of three independent experiments.

**Figure 7 cancers-15-00720-f007:**
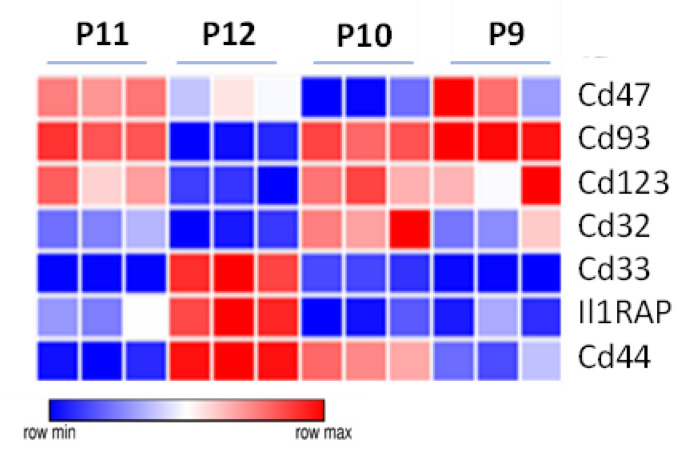
Specific drug-targeted genes expressed in ML23 cells. Differential expression levels of current treatment targets that are under study or clinical application were examined in the RNA-seq results of ML23 cells. Two of them (CD47 and CD93) presented overexpression on ML23 LIC (P11).

## Data Availability

The complete RNA seq data had been deposited GSE220908.

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
