# Peer review of "Molecular Mechanisms in Murine Syngeneic Leukemia Stem Cells"

_cancers, 2023, doi:10.3390/cancers15030720_

Round 1
Reviewer 1 Report
The authors present heterogeneity in murine AML ML23 line which they established previously. They now show that an isolated sub population of cells defined by CD9-/c-Kit+/Mac1-/CD48+ when injected in immunocompetent mice, engrafts at all primary and secondary hematopoietic sites while giving rise to CD9+ and Mac1+ cells as well.
The study has substantial potential and can be presented better. It will be a good idea to explain the sorted populations by their two or four marker combination instead of writing P9, 10, 11, and 12, where it is expected from the reader to either remember the populations from figure 1 or keep going back to it. Figure resolution can be improved and a bar graph and couple of "heat maps" can be merged. The sentence in line 50 is incorrect. As per figure 1, majority of c-kit positive cells are ZsGreen dim to negative. It is unclear that when the cells were sorted based on c-kit and CD9 combination for figure 2, were they pre gated on ZsGreen or not? If cells were not pre gated, then P10 was most likely normal HSCs. Figure 3 and its text is unclear, while in 3a the cells before injection are shown, in the text it sounds like these are the cells extracted after injection. Did authors perform secondary transplants from the engrafted P11?. In figure 4, did authors expect a difference within the bone marrow coming from femur, tibia and pelvis, if so then what was the reason? A PCA plot in figure5 comparing distance between populations would have made it easier to the point about transcriptional proximity of P11 and P12.
Author Response
Reviewer#1 comments and responses
The authors present heterogeneity in murine AML ML23 line which they established previously. They now show that an isolated sub population of cells defined by CD9-/c-Kit+/Mac1-/CD48+ when injected in immunocompetent mice, engrafts at all primary and secondary hematopoietic sites while giving rise to CD9+ and Mac1+ cells as well.
The study has substantial potential and can be presented better.
1.1 It will be a good idea to explain the sorted populations by their two or four marker combination instead of writing P9, 10, 11, and 12, where it is expected from the reader to either remember the populations from figure 1 or keep going back to it.
Thanks for this good comment. We had added explicit explanation of the markers in the text. Page 6 lines 150-152, and in Figure 5. This indeed will help the reader.
1.2 Figure resolution can be improved and a bar graph and couple of "heat maps" can be merged.
Thanks for this note. We improved resolution of figures, and can upload figures at even higher resolution as requested.
1.3 The sentence in line 50 is incorrect. As per figure 1, majority of c-kit positive cells are ZsGreen dim to negative. It is unclear that when the cells were sorted based on c-kit and CD9 combination for figure 2, were they pre-gated on ZsGreen or not? If cells were not pre gated, then P10 was most likely normal HSCs.
Thanks for this comment, reading again the manuscript we agree with this remark. We deleted this sentence.
As per sorting of the cells, they were pre-gated ZsGreen+, this is now clearly stated in the text and figure. Page 6 line 151, and Figure 2.
1.4 Figure 3 and its text is unclear, while in 3a the cells before injection are shown, in the text it sounds like these are the cells extracted after injection. Did authors perform secondary transplants from the engrafted P11?
Thanks for this accurate comment. Figure 3a is showing the cells before injection (as sorted from donor mice). Figure 3b is showing the cells after injection, from the recipient of P11-sorted cells (not secondary, just the recipient of the P11 sub-population). This is now clearly stated in the text, page 7 line 165-169, and in Figure 3.
1.5 In figure 4, did authors expect a difference within the bone marrow coming from femur, tibia and pelvis, if so then what was the reason?
Thanks for this smart note. Indeed, as reviewer suggest, we did not expect a difference. Nevertheless, it was technically possible and we were curious to realize if indeed, as expected, there is no difference. Since we did this extra-effort, we do show the data, just in case any of the readers might be curious about it too. This is now added to the text, page 7 lines 180-182.
1.6 A PCA plot in figure5 comparing distance between populations would have made it easier to the point about transcriptional proximity of P11 and P12.
Thanks for this suggestion. We add a PCA plot to Figure 5, and we thank the reviewer for helping us to better visualize our data. Indeed, as reviewer anticipated, P12 is the most proximal to P11.
Reviewer 2 Report
Michael Chamo and collegues present a high quality and well-written experimental manuscript focused on molecular mechanisms in murine syngeneic leukemia stem cells.
Authors present prospective isolation of a defined LSC sub-population, encompassing the potency to passage disease from mouse to mouse. They further provide molecular insights of whole transcriptome analysis. Importantly, the ML23 LSC sub-population is expressing therapeutic targeted genes and provides a model for research in immune-competent animals.
Authors searched for possible LSCs in immunocompetent synergetic mice model. First, they found phenotypic heterogeneity in the ML23 leukemia line. They prospectively isolate a sub-population with the surface markers cKit+CD9-CD48+Mac1-/low that have the potency to relapse the disease. Importantly, this sub-population can passage in syngeneic hosts, and retrieve the heterogeneity of the parental ML23 leukemia line. The LSC sub-population resides in various organs. Then they present a unique gene expression signature of the LSC in the ML23 model compared to the other sub-populations. Interestingly, the ML23 LSC sub-population expresses therapeutic targeted genes such as CD47 and CD93. Taken together, authors present the identification and molecular characterization of LSCs in a syngeneic murine model.
Finally, authors conclude that their data provides novel models for LSC in syngeneic mice. This, and additional models, will improve the understanding of heterogeneous AML. Studying multiple syngeneic leukemic mice may resolve current controversies and most likely realize variability which relevant to differences between human patients.
Overall, the manuscript is highly valuable for the scientific community and should be accepted for publication.
======================
Other comments to authors:
1) Please check for typos throughout the manuscript.
2) With regards to acute lymphoblastic leukemia (ALL) – authors are kindly encouraged to cite the following article that describes mechanisms of T cell therapy dysfunction. DOI: 10.3390/cancers14041078
Author Response
Reviewer#2 comments and responses
Michael Chamo and collegues present a high quality and well-written experimental manuscript focused on molecular mechanisms in murine syngeneic leukemia stem cells.
Authors present prospective isolation of a defined LSC sub-population, encompassing the potency to passage disease from mouse to mouse. They further provide molecular insights of whole transcriptome analysis. Importantly, the ML23 LSC sub-population is expressing therapeutic targeted genes and provides a model for research in immune-competent animals.
Authors searched for possible LSCs in immunocompetent synergetic mice model. First, they found phenotypic heterogeneity in the ML23 leukemia line. They prospectively isolate a sub-population with the surface markers cKit+CD9-CD48+Mac1-/low that have the potency to relapse the disease. Importantly, this sub-population can passage in syngeneic hosts, and retrieve the heterogeneity of the parental ML23 leukemia line. The LSC sub-population resides in various organs. Then they present a unique gene expression signature of the LSC in the ML23 model compared to the other sub-populations. Interestingly, the ML23 LSC sub-population expresses therapeutic targeted genes such as CD47 and CD93. Taken together, authors present the identification and molecular characterization of LSCs in a syngeneic murine model.
Finally, authors conclude that their data provides novel models for LSC in syngeneic mice. This, and additional models, will improve the understanding of heterogeneous AML. Studying multiple syngeneic leukemic mice may resolve current controversies and most likely realize variability which relevant to differences between human patients.
Overall, the manuscript is highly valuable for the scientific community and should be accepted for publication.
======================
Other comments to authors:
2.1 Please check for typos throughout the manuscript.
Thanks for this comment; we checked the manuscript and reference list and corrected typos.
2.2 With regards to acute lymphoblastic leukemia (ALL) – authors are kindly encouraged to cite the following article that describes mechanisms of T cell therapy dysfunction. DOI: 10.3390/cancers14041078
Thanks for this good reference; we added it to the introduction and the discussion (Reference # 20).
Round 2
Reviewer 1 Report
Thank you for the notes, revised remarks are satisfactory. Congratulations on the accomplished work.
Reviewer 2 Report
Corrections have been made. The manuscript can be accepted.